# NetworkGym: Reinforcement Learning Environments for Multi-Access Traffic Management in Network Simulation

**Momin Haider**[*]
UC, Santa Barbara

**Ming Yin**[†]
Princeton University

**Menglei Zhang**
Intel Labs

**Arpit Gupta**
UC, Santa Barbara

**Jing Zhu**
Intel Labs

**Yu-Xiang Wang**
UC, San Diego

## Abstract

Mobile devices such as smartphones, laptops, and tablets can often connect to multiple access networks (e.g., Wi-Fi, LTE, and 5G) simultaneously. Recent advancements facilitate seamless integration of these connections below the transport layer, enhancing the experience for apps that lack inherent multi-path support. This optimization hinges on dynamically determining the traffic distribution across networks for each device, a process referred to as *multi-access traffic splitting*. This paper introduces *NetworkGym*, a high-fidelity network environment simulator that facilitates generating multiple network traffic flows and multi-access traffic splitting. This simulator facilitates training and evaluating different RL-based solutions for the multi-access traffic splitting problem. Our initial explorations demonstrate that the majority of existing state-of-the-art offline RL algorithms (e.g. CQL) fail to outperform certain hand-crafted heuristic policies on average. This illustrates the urgent need to evaluate offline RL algorithms against a broader range of benchmarks, rather than relying solely on popular ones such as D4RL. We also propose an extension to the TD3+BC algorithm, named Pessimistic TD3 (PTD3), and demonstrate that it outperforms many state-of-the-art offline RL algorithms. PTD3's behavioral constraint mechanism, which relies on value-function pessimism, is theoretically motivated and relatively simple to implement.

## 1   Introduction

There exists a general lack of standardized benchmarks for reinforcement learning (RL) in the domain of computer networking. Whereas RL has shown promise in addressing various challenges in computer networking, such as congestion control, routing, and resource allocation, the field lacks widely accepted benchmarks that would facilitate systematic evaluation and comparison of different RL approaches. Hence, we propose *NetworkGym*, a high-fidelity, end-to-end, full-stack network Simulation-as-a-Service framework that leverages open-source network simulation tools, such as ns-3 Henderson et al. [2008]. Furthermore, NetworkGym offers a closed-loop machine learning (ML) algorithm development and training pipeline via open-source gym-like APIs. The components of NetworkGym achieve the following objectives:

---

[*]Rest in peace.

[†]Most of the work was done while Ming Yin was at UCSB. Correspondence to: `my0049@princeton.edu`, `jing.zhu.ietf@gmail.com`, and `yuxiangw@ucsd.edu`.

38th Conference on Neural Information Processing Systems (NeurIPS 2024) Track on Datasets and Benchmarks.

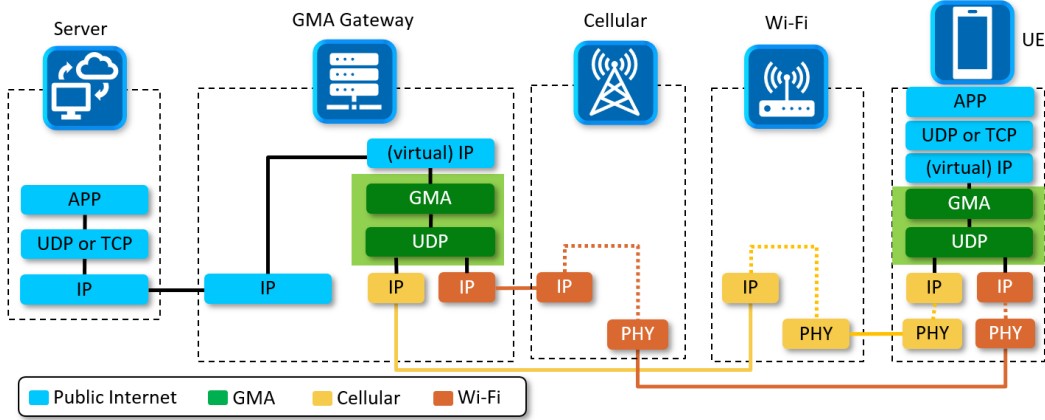

Figure 1: GMA Protocol. A UE interfaces with the GMA gateway over UDP. "APP" refers to the application layer at the client or server level, "IP" refers to the Internet Protocol layer, facilitating the addressing and routing of packets, and "PHY" refers to the physical layer in the network responsible for the actual transmission of data over the network medium. The GMA gateway handles multi-access traffic splitting at the edge.

- **Open APIs for ML Training and Data Collection**: The Agent is fully customizable and controlled by the developer. The network simulation Environment is hosted in the cloud. By utilizing the open-source NetworkGym Client and APIs, an Agent can interact with an Environment to collect measurement data and take actions that allow training for the desired use case.

- **Flexibility of Programming Language**: The separation of Agent and Environment provides the freedom to employ different programming languages for the ML algorithm and network simulation. For instance, a Python-based Agent can smoothly interact with a C++ (ns-3) based simulation Environment. This is a critical aspect of our framework, as previous networking frameworks would have required modern ML algorithms to be coded in the same language(s) as the simulation environment.

- **Independent and Modular Deployment**: Such separation also allows the Agent and Environment to be deployed on different machines or platforms, optimized for specific workloads. For example, when training online on-policy algorithms, such as PPO Schulman et al. [2017] and SAC Haarnoja et al. [2018a,b], it is often critical to parallelize environment instances to accelerate training and improve generalization capability Wijmans et al. [2019], Makoviychuk et al. [2021]. This would be difficult to accomplish if the Agent and Environment were coupled. They can also be developed and maintained by different entities. Access to the Environment is controlled through NetworkGym APIs to hide the details of how a network function or feature is implemented from developers.

**Motivation from Computer Networking.** As the mobile industry evolves toward 6G, it is becoming clear that no single access technology will be able to meet the great variety of requirements for human and machine communications. Multi-access traffic management for integrating multiple heterogeneous wireless networks, e.g., Wi-Fi, cellular, satellite, etc., into a virtualized and unified network becomes vital for addressing today's ever-increasing performance requirements and future applications. Recently, the Generic Multi-Access (GMA) protocol has been proposed in the Internet Engineering Task Force (IETF) to address this need Zhu and Zhang [2024], and the 3rd Generation Partnership Project (3GPP) has also developed the access traffic steering, switching, and splitting (ATSSS) feature, which enables simultaneous use of one 3GPP and one non-3GPP connection to deliver data flows ats [2018]. We defer more technical details of these protocols to Appendix A.

One effective method for managing multi-access traffic is through traffic splitting between different network types. Specifically, for each user equipment (UE), traffic is allocated between a 3GPP connection (e.g., LTE) and a non-3GPP connection (e.g., Wi-Fi), with the ratio adjusted at frequent intervals, as in Figure 1. It is natural to consider using RL for learning adaptive and data-driven decision policies on the traffic-splitting ratios.

Applying RL, however, is notoriously hard. One may run *online RL* on real networks, but the initial decisions made by the algorithms can be suboptimal, leading to poor network traffic splitting and diminished user experience. Notably, in applications such as robotic control over networks, it is critical to ensure high reliability and low packet-loss ratios to maintain operational effectiveness. An alternative is to run *offline RL* on the logged data from real networks, but data coverage is a big issue. Even if the learned policy from offline RL improves over the baseline, one cannot know for sure until testing it online with real network traffic. Moreover, the networking environment is not static and most challenging scenarios occur in the long tail of the data distribution.

NetworkGym is timely as it allows us to not only evaluate any learned RL policies, but also stress-test them in challenging scenarios. One could also use NetworkGym to simulate the entire workflow of offline RL for policy improvement before deploying the workflow on real networks.

**Frictionless Reproducibility for ML Researchers interested in Computer Networking.** The intended use of NetworkGym is to allow machine learning (especially RL) researchers to evaluate their algorithms on a faithfully simulated environment in computer networking without having to understand the intricate networking protocols and their interactions in a multi-access traffic splitting system. To facilitate "frictionless reproducibility" Donoho [2024], we conduct preliminary experiments on NetworkGym with popular offline RL algorithms and make the code to setup such experiments available. Our results provide the following take-home messages:

- **Offline RL for Policy Improvement.** Offline RL algorithms can effectively improve the performance in networking systems using data collected from three behavioral policies.
- **Transferability of Scientific Advances.** Methods that work well on standard OpenAI gym environments may not work well on networking problems. Comparative advantages of State-of-the-Art algorithms on D4RL Fu et al. [2020] do not *transfer* to NetworkGym.
- **Details matter.** Seemingly arbitrary choices in the parameterization and state/action representation (e.g., normalization) have more substantial impact than the choice of RL algorithms.
- **Success of principles.** "Pessimism" in offline RL works for networking problems. A more theory-inspired pessimistic bonus is more effective than the popular Behavioral Cloning (BC).

We hope NetworkGym lowers the entry-barrier into computer networking research and enables new collaboration in the emerging research area of machine learning for networking across academia and industry.

## 2 Related Work

**RL-based Network Optimization.** RL has been used for network optimization in a variety of contexts Yang et al. [2024], Mao et al. [2017], Jay et al. [2019], Jamil et al. [2022], Zhang et al. [2023], Xia et al. [2022], Gilad et al. [2019], Boyan and Littman [1993], Wei et al. [2022], He et al. [2017], Liang et al. [2019], Sadeghi et al. [2017]. For example, Yang et al. Yang et al. [2024] use offline RL on a mixture of datasets from different behavior policies to maximize throughput via radio resource management. Additionally, Mao et al. Mao et al. [2017] construct a system that can generate adaptive bitrate algorithms to maximize user quality of experience by training a deep RL model on client video player observations. Jay et al. Jay et al. [2019] employ deep RL to solve the congestion control problem, whereas Jamil et al. Jamil et al. [2022] use deep RL to dynamically determine the optimal number of TCP streams in parallel to maximize throughput while avoiding network congestion. Despite the existing works, our use of offline RL for multi-access traffic splitting is novel and the first of its kind.

**RL Benchmarks.** A wide variety of online and offline RL benchmarks have been proposed in the research community in order to properly evaluate the performance and generalization of RL algorithms. Popular online RL benchmarks include the OpenAI Gym Brockman et al. [2016], Atari 2600 games bel [2013], and Mujoco Todorov et al. [2012]. These sets of environments offer a diverse selection of tasks to choose from, mostly involving classic control, continuous control of multi-joint bodies, or video game playing with high-dimensional input spaces. Common offline RL benchmarks include D4RL Fu et al. [2020] and RL Unplugged Gulcehre et al. [2020], which provide similar environments to those in the referenced online RL benchmarks. Recent efforts have been made to consolidate these offline RL benchmarks and have also reinforced the finding that

the success of offline RL methods strongly depends on the training data distribution Kang et al. [2023]. Additionally, Voloshin et al. Voloshin et al. [2019] introduce the COBS off-policy evaluation benchmarking suite to comprise a much wider variety of environments than simply the Mujoco-style or Atari-style ones. However, none of these benchmarks contains environments that focus on computer networking applications.

**Offline RL.** Most approaches to offline RL involve some form of behavioral constraint or policy regularization to ensure that the actions chosen by the policy don't stray too far from the actions in the dataset for corresponding states Levine et al. [2020], Kostrikov et al. [2021b], Kumar et al. [2020], Kostrikov et al. [2021a], Yin and Wang [2021], Li et al. [2023]. This is used to mitigate the distribution shift between training and testing states. Certain algorithms seek to avoid off-policy evaluation (OPE) altogether, due to the inherent associated high variance which is compounded on each training iteration Brandfonbrener et al. [2021]. Other algorithms use a form of divergence constraint to control the resulting behavior policy. For example, Conservative $Q$-Learning (CQL) modifies the actor-critic framework by selecting a policy whose expected value under a $Q$-function lower-bounds its true value Kumar et al. [2020]. Implicit $Q$-Learning (IQL) seeks to avoid policy evaluation on unseen actions and instead treats the state value function as a random variable; the value of the best actions at a state can then be estimated by taking the upper expectile of the value function conditioned on the state Kostrikov et al. [2021b].

**Offline RL Using Online Algorithms.** Other offline RL methods take advantage of the empirical success of state-of-the-art online RL methods; we include a discussion of some of these algorithms in Appendix B. Additionally, Fujimoto et al. Fujimoto and Gu [2021] propose a minimal extension to the popular online RL algorithm TD3 by augmenting the policy improvement step with a simple behavioral cloning term. We note that while behavioral cloning is one way to prevent learned policies from excessively favoring out-of-distribution (OOD) actions, another possibility is to incorporate some form of pessimism into the $Q$-value estimates for these OOD actions. In particular, our work is inspired by that of Yin et al. Yin et al. [2023] in which the authors analyze the Pessimistic Fitted $Q$-Learning (PFQL) algorithm and show that in the finite-horizon case, it is provably sample efficient under certain assumptions. Their approach involves computing the Fisher information matrix on the offline dataset with respect to the $Q$-function approximator and using that matrix to estimate the uncertainty of any state-action pair. In this way, they are able to compute policies that maximize a lower-bound estimate of the state-action value function, improving the performance of the algorithm in the offline RL setting. In our work, we incorporate this idea of introducing pessimism into the $Q$-value estimates of TD3 in a similar way that Fujimoto et al. introduce behavioral cloning to TD3. Specifically, we adjust the policy improvement step to account for uncertainties present in the $Q$-values of specific state-action pairs and produce a resulting algorithm we denote as Pessimistic TD3 (PTD3). We introduce PTD3 in Appendix C.

## 3   Problem Setup

**Markov Decision processes.** Let $(\mathcal{S}, \mathcal{A}, \mathcal{R}, p, \gamma)$ define a Markov decision process (MDP) where $\mathcal{S}$ is the state space, $\mathcal{A}$ is the action space, $\mathcal{R} : \mathcal{S} \times \mathcal{A} \to \mathbb{R}$ is the scalar reward function, $p : \mathcal{S} \times \mathcal{A} \to \Delta^{\mathcal{S}}$ is the transition dynamics model where $\Delta^{\mathcal{S}}$ is a set of probability distributions over $\mathcal{S}$, and $\gamma \in [0, 1]$ is the discount factor. $\mathcal{S}$ and $\mathcal{A}$ can both potentially be infinite or continuous. Typically, an RL agent chooses actions via a deterministic policy $\mu : \mathcal{S} \to \mathcal{A}$ or a stochastic policy $\pi : \mathcal{S} \to \Delta^{\mathcal{A}}$ where $\Delta^{\mathcal{A}}$ is a set of probability distributions over $\mathcal{A}$. The goal of an RL agent is to find a policy that maximizes the expected discounted return $\mathbb{E}_{\pi} [\sum_{t=0}^{\infty} \gamma^t r_t | s_0 = s]$ from the starting state distribution. We denote the state-action value function with respect to policy $\pi$ as $Q^{\pi}(s, a) = \mathbb{E}_{\pi} [\sum_{t=0}^{\infty} \gamma^t r_t | s_0 = s, a_0 = a]$.

**Offline RL.** In the offline RL setting, we assume that the agent does not have the ability to interact with the environment and instead has access to an offline dataset $\mathcal{D} = \{(s_k, a_k, r_k, s'_k)\}_{k=1}^{K}$ collected by some unknown data-generating process (for example, a collection of different behavior policies). This makes the offline RL setting more challenging than the online RL setting. An online RL agent that overestimates the $Q$-values at specific state-action pairs can quickly adapt after being punished for taking those actions in the environment, but an offline RL agent does not have the ability to interact with the environment. This leads to the resulting problem of distribution shift in offline RL, which occurs due to extrapolation error in the $Q$-function approximators on state-action pairs that are poorly represented by those in the offline dataset.

**Multi-Access Traffic Splitting Environment.** In the Multi-Access Traffic Splitting environment, a predetermined number of UEs are randomly distributed on a 2-dimensional grid. When the environment is first instantiated, each UE is connected to a single LTE base station and the nearest Wi-Fi access point. The location range of the UEs and the locations of the base station and access points may be specified at environment initialization. If the RSSI-based handover is enabled in the NetworkGym environment configuration, then the Wi-Fi access point for each UE will dynamically change during the simulation to whichever has the highest received signal. Each time step in the environment consists of a time interval of 0.1 seconds. During this time interval, traffic-related measurements are taken, such as the one-way-delay and output traffic throughput. The goal of a centralized traffic splitting agent is to strategically split traffic over the Wi-Fi and LTE links, aiming to achieve high throughput and low latency. Within the NetworkGym environment configuration, it is possible to specify parameters that control the nature of the UEs' movement and whether or not they follow a random or deterministic walk.

**Observation Space.** An observation at time $t$ is $s(t) = [s_1(t), s_2(t), ..., s_{N_u}(t)]$ for $N_u$ users where $s_j(t)$ is a tuple of values for the $j$-th UE in the following form: $(lc_{\text{LTE}}, lc_{\text{Wi-Fi}}, tp_{\text{in}}, tp_{\text{out, LTE}}, tp_{\text{out, Wi-Fi}}, owd_{\text{LTE}}, owd_{\text{Wi-Fi}}, owd_{\text{max, LTE}}, owd_{\text{max, Wi-Fi}}, id_{\text{Wi-Fi}}, sr_{\text{LTE}}, sr_{\text{Wi-Fi}}, x, y)$, $lc_{\text{k}}$ is the UE's link capacity for channel k, $tp_{\text{in}}$ is the UE's input traffic throughput, $tp_{\text{out, k}}$ is the UE's output traffic throughput across channel k, $owd_{\text{k}}$ is the UE's one-way-delay across channel k, $owd_{\text{max, k}}$ is the UE's maximum one-way-delay across channel k, $id_{\text{Wi-Fi}}$ is the UE's current Wi-Fi access point ID, $sr_{\text{k}}$ is the UE's splitting ratio for channel k, $x$ is the x-location of the user, and $y$ is the y-location of the user.

**Action Space.** An action at time $t$ takes the form $a(t) = [a_1(t), a_2(t), ..., a_{N_u}(t)]$ for $N_u$ users where $a_j(t) \in \left\{ \frac{0}{32}, \frac{1}{32}, ..., \frac{31}{32}, \frac{32}{32} \right\}$ is the desired Wi-Fi splitting ratio for the $j$-th UE during the next time interval.

**Reward Function.** The immediate reward at time $t$ is computed as in Equation 1 where $tp_i$ and $dy_i$ are the output traffic throughput and one-way-delay across both channels for the $i$-th user during the current time interval, respectively. Furthermore, $tp_{i,\text{max}}$ is the sum of link capacities across both channels for the $i$-th user during this time interval and we take $dy_{\text{max}}$ to be 1000ms, after which a packet is treated as lost. In this way, we normalize the reward function to be invariant to unit-translation and incentivize a learning agent to maximize the average throughput while simultaneously minimizing the average delay across channels. Although this reward function is admittedly somewhat arbitrary, a different reward function can be easily specified by the network administrators in order to satisfy different QoS requirements.

$$r(t) = \log \left( \frac{1}{N_u} \sum_{i=1}^{N_u} \frac{tp_i}{tp_{i,\text{max}}} \right) - \log \left( \frac{1}{N_u} \sum_{i=1}^{N_u} \frac{dy_i}{dy_{\text{max}}} \right) \tag{1}$$

## 4 Experiments

**Experimental Setup.** We test PTD3 and other state-of-the-art offline RL algorithms on a simplified configuration of the NetworkGym multi-access traffic splitting environment. The relevant environment configuration file is included in Appendix D. At initialization of each environment, four UEs are randomly stationed 1.5 meters above the $x$-axis between $x = 0$ and $x = 80$ meters. From there, they begin to bounce back and forth in the $x$-direction at 1 m/s for the entire duration of an episode. The Wi-Fi access points are stationed at $(x, z) = (30\text{m}, 3\text{m})$ and $(x, z) = (50\text{m}, 3\text{m})$, respectively while the LTE base station lies at $(x, z) = (40\text{m}, 3\text{m})$. Figure 2 illustrates this environment setting. Although this setup is deceptively simple and unrealistic due to the relative locations between UEs and access points as well as the degenerate movement of the UEs, it provides a simple enough testing ground for offline RL on the GMA traffic splitting protocol while still containing some amount of dynamic behavior and resource competition between UEs.

Since the multi-access gateway is connected to all four UEs, the gateway can send traffic splitting command messages to each of the UEs in a centralized manner via the GMA protocol while taking into account network information across all UEs. Therefore, we represent the state as a $14 \times 4$ matrix where we have 14 network measurement values for each user from the previous time interval and we represent the action as a $1 \times 4$ row-vector, where each element represents the desired traffic splitting

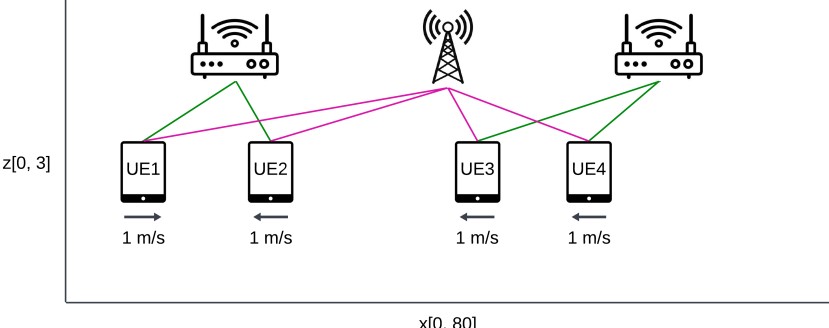

Figure 2: Environment configuration for offline RL testing (not-to-scale). Here, we randomly initialize four UE's 1.5 meters above the $x$-axis and they move back and forth in the $x$-direction between $x = 0$ meters and $x = 80$ meters. The Wi-Fi access point locations are $(x, z) = (30\text{m}, 3\text{m})$ and $(x, z) = (50\text{m}, 3\text{m})$ while the LTE base station location is $(x, z) = (40\text{m}, 3\text{m})$.

ratio during the next time interval for a specific user. Although the traffic splitting ratio for each user can only be one of 33 discrete values $\left(\frac{0}{0}, \frac{1}{32}, ..., \frac{32}{32}\right)$, we treat each element in the action as a continuous real number between 0 and 1 and map it to the closest corresponding discrete value.

**Heuristic Policies.** NetworkGym provides three heuristic policies for traffic splitting and offline data collection, which we denote `throughput_argmax`, `system_default`, and `utility_logistic`. All of these policies operate independently on each UE, without considering coupled interactions between them. For each user, `throughput_argmax` examines the previous Wi-Fi and LTE link capacities and chooses the traffic splitting ratio to completely favor whichever channel previously had the highest link capacity. For the `system_default` algorithm, if the UE-specific difference in delay among the Wi-Fi and LTE links exceeds a threshold, traffic over the link with lower delay is gradually increased. If the delay difference among both links is small but packet loss is detected, traffic over the link with a lower packet loss rate is incrementally increased. The final heuristic policy, `utility_logistic`, computes a Wi-Fi utility $u_{i,\text{Wi-Fi}} = \log\left(1 + tp_{i,\text{Wi-Fi}}\right) - \log\left(1 + dy_{i,\text{Wi-Fi}}\right)$ and the corresponding LTE utility $u_{i,\text{LTE}}$ for each user and then computes the desired traffic splitting ratio for said user as $\sigma\left(u_{i,\text{Wi-Fi}} - u_{i,\text{LTE}}\right)$ where $\sigma(\cdot)$ is the logistic function. In this way, the traffic splitting ratio favors channels that indicated higher utility during the previous time interval.

**Offline Datasets.** For each heuristic policy, we collect an offline dataset over 64 episodes, each with a different starting configuaration of UEs. Each episode consists of 10,000 steps. We evaluate the offline dataset coverages for each algorithm by computing the minimum eigenvalue of the feature covariance matrix $\mathbf{C} = \mathbb{E}_{s,a\sim\mathcal{D}}\left[\phi(s,a)\phi(s,a)^{\text{T}}\right]$ where $\phi(s,a)$ is the featurization of the state-action pair Jin et al. [2020, 2021], Zanette et al. [2021], Yin et al. [2022], Nguyen-Tang et al. [2023]. We featurize state-action pairs by simply concatenating the flattened state and action vectors together.

The minimum eigenvalue and condition number for each population covariance matrix are illustrated in Table 1. The offline dataset coverage is highest for the `utility_logistic` algorithm and lowest for the `throughput_argmax` algorithm. In practice, the `throughput_argmax` algorithm sends all traffic through the Wi-Fi channel over 99% of the time for each user, with occasional bursts over LTE while the `utility_logistic` algorithm thrashes back and forth for each user between sending traffic over Wi-Fi and LTE, resulting in a dataset with much higher coverage. The results in Table 1 present a wide range of dataset coverage values, spanning multiple orders of magnitude. This diverse set of benchmarks ensures that offline RL algorithms can be appropriately evaluated. For instance, algorithms trained on datasets with low coverage are expected to adhere closely to the behavior policy due to limited data variety, while those trained on high-coverage datasets have the potential for greater improvement over the behavior policy due to more diverse experiences to learn from.

In Table 3, we evaluate the performance of three heuristic policies, several offline RL algorithms trained on different datasets, and two state-of-the-art online RL algorithms (PPO and SAC) in this environment setting. The online RL algorithms establish a soft upper bound on the returns achievable by offline RL algorithms in our NetworkGym environment setting. For each of the algorithms, we evaluate its performance on 40 evaluation episodes, each of which is 3200 steps. The total return

| dataset-generating algorithm | $\lambda_{\min}(\mathbf{C})$ | $\kappa(\mathbf{C})$ |
|---|---|---|
| **throughput_argmax** | $4.4 \cdot 10^{-6}$ | 4,949,213 |
| **system_default** | $1.5 \cdot 10^{-4}$ | 220,682 |
| **utility_logistic** | $4.8 \cdot 10^{-4}$ | 34,827 |

Table 1: Offline Dataset Coverage Measurements among Heuristic Policies. $\lambda_{\min}(\mathbf{C})$ and $\kappa(\mathbf{C})$ are the minimum eigenvalue and condition number of the feature covariance matrix, respectively.

per step across all episodes is then averaged and reported; the error bar indicates a 95% confidence interval centered around the mean. The performance across different datasets is then averaged again to produce an average performance across all datasets in the rightmost column. Finally, in Table 5, we examine the performance of the PTD3 algorithm across the different datasets and different values of $\beta$ where $\alpha = 1.0$. We find that setting $\alpha = 1.0$ results in the least variance in performance across values of $\beta$.

| | thrpt_argmax | system_default | utility_logistic | Average |
|---|---|---|---|---|
| **baseline** | $0.747 \pm 0.049$ | $0.555 \pm 0.052$ | $0.949 \pm 0.039$ | $0.750 \pm 0.047$ |
| **BC (norm)** | $0.749 \pm 0.047$ | $0.825 \pm 0.042$ | $0.749 \pm 0.047$ | $0.774 \pm 0.045$ |
| **BC (no norm)** | $0.751 \pm 0.049$ | $0.433 \pm 0.054$ | $0.946 \pm 0.039$ | $0.710 \pm 0.047$ |
| **CQL (norm)** | $0.749 \pm 0.047$ | $0.749 \pm 0.047$ | $0.749 \pm 0.047$ | $0.749 \pm 0.047$ |
| **CQL (no norm)** | $\mathbf{0.998 \pm 0.043}$ | $0.381 \pm 0.082$ | $0.957 \pm 0.044$ | $0.779 \pm 0.056$ |
| **IQL (norm)** | $0.770 \pm 0.051$ | $0.818 \pm 0.042$ | $0.748 \pm 0.048$ | $0.779 \pm 0.047$ |
| **IQL (no norm)** | $0.749 \pm 0.049$ | $0.846 \pm 0.042$ | $0.948 \pm 0.036$ | $0.848 \pm 0.042$ |
| **TD3+BC (norm)** | $0.749 \pm 0.047$ | $0.034 \pm 0.046$ | $0.749 \pm 0.047$ | $0.511 \pm 0.047$ |
| **TD3+BC (no norm)** | $0.778 \pm 0.047$ | $0.906 \pm 0.049$ | $0.863 \pm 0.038$ | $0.849 \pm 0.045$ |
| **EDAC** | $0.336 \pm 0.285$ | $-0.888 \pm 0.034$ | $0.913 \pm 0.027$ | $0.120 \pm 0.115$ |
| **LB-SAC** | $0.902 \pm 0.046$ | $-0.204 \pm 0.072$ | $\mathbf{1.150 \pm 0.033}$ | $0.616 \pm 0.050$ |
| **SAC-N** | $0.838 \pm 0.052$ | $0.817 \pm 0.035$ | $0.699 \pm 0.026$ | $0.785 \pm 0.038$ |
| **PTD3** | $0.746 \pm 0.050$ | $\mathbf{1.013 \pm 0.039}$ | $\mathbf{1.079 \pm 0.040}$ | $\mathbf{0.946 \pm 0.043}$ |
| **PPO** | $\sim$ | $\sim$ | $\sim$ | $1.214 \pm 0.037$ |
| **SAC** | $\sim$ | $\sim$ | $\sim$ | $1.104 \pm 0.037$ |

Table 2: Offline and Online RL Algorithm Performance Across Multiple Offline Datasets. Each of the first three column headers indicates the baseline algorithm that collected the offline dataset where "thrpt_argmax" is an alias for throughput_argmax. Each row header (except "baseline", "PPO", and "SAC") is an offline RL algorithm trained on one of three offline datasets. "baseline" refers to the performance of the original baseline heuristic policies without any offline data collection. "(norm)" indicates that the algorithm implements state-normalization based on the offline dataset while "(no norm)" indicates that the algorithm does not. If not specified, the algorithm does not implement state normalization. We use $\alpha = 1.0$ and $\beta = 10.0$ in our evaluation of PTD3.

## 5 Discussion

**Offline RL Algorithm Performance.** First, we note that of the 7 off-the-shelf offline RL algorithms tested in our NetworkGym environment setting, only 2 of them were able to significantly outperform the average performance of the heuristic baseline algorithms. Furthermore, in the case of both of these algorithms, they were only able to do so when we disabled state normalization based on the offline dataset, a feature that is included by default when training these offline algorithms. Therefore, using the default hyperparameters for every tested off-the-shelf offline RL algorithm, *none* of these algorithms could significantly outperform the heuristic baseline algorithms on average. Furthermore, in the case of a few of these algorithms, such as EDAC and LB-SAC, the performance across different datasets is erratic, resulting in a significantly *lower* average performance overall, compared to the heuristic baseline algorithms. While these algorithms are known to exhibit state-of-the-art performance on D4RL-like tasks, it has been noted that the performance of these algorithms in practice is unstable across environments of varying characteristics Tarasov et al. [2024]. These

|  | thrpt_default | delay_default | utility_default | Average |
|---|---|---|---|---|
| **baseline** | $0.747 \pm 0.049$ | $0.555 \pm 0.052$ | $0.949 \pm 0.039$ | $0.750 \pm 0.047$ |
| **BC (norm)** | $0.749 \pm 0.047$ | $0.825 \pm 0.042$ | $0.749 \pm 0.047$ | $0.774 \pm 0.045$ |
| **BC (no norm)** | $0.751 \pm 0.049$ | $0.433 \pm 0.054$ | $0.946 \pm 0.039$ | $0.710 \pm 0.047$ |
| **CQL (norm)** | $0.749 \pm 0.047$ | $0.749 \pm 0.047$ | $0.749 \pm 0.047$ | $0.749 \pm 0.047$ |
| **CQL (no norm)** | $\mathbf{0.998 \pm 0.043}$ | $0.381 \pm 0.082$ | $0.957 \pm 0.044$ | $0.779 \pm 0.056$ |
| **IQL (norm)** | $0.770 \pm 0.051$ | $0.818 \pm 0.042$ | $0.748 \pm 0.048$ | $0.779 \pm 0.047$ |
| **IQL (no norm)** | $0.749 \pm 0.049$ | $0.846 \pm 0.042$ | $0.948 \pm 0.036$ | $0.848 \pm 0.042$ |
| **TD3+BC (norm)** | $0.749 \pm 0.047$ | $0.034 \pm 0.046$ | $0.749 \pm 0.047$ | $0.511 \pm 0.047$ |
| **TD3+BC (no norm)** | $0.778 \pm 0.047$ | $0.906 \pm 0.049$ | $0.863 \pm 0.038$ | $0.849 \pm 0.045$ |
| **EDAC** | $0.336 \pm 0.285$ | $-0.888 \pm 0.034$ | $0.913 \pm 0.027$ | $0.120 \pm 0.115$ |
| **LB-SAC** | $0.902 \pm 0.046$ | $-0.204 \pm 0.072$ | $\mathbf{1.150 \pm 0.033}$ | $0.616 \pm 0.050$ |
| **SAC-N** | $0.838 \pm 0.052$ | $0.817 \pm 0.035$ | $0.699 \pm 0.026$ | $0.785 \pm 0.038$ |
| **PTD3** | $0.746 \pm 0.050$ | $\mathbf{1.013 \pm 0.039}$ | $\mathbf{1.079 \pm 0.040}$ | $\mathbf{0.946 \pm 0.043}$ |
| **PPO** | $\sim$ | $\sim$ | $\sim$ | $1.214 \pm 0.037$ |
| **SAC** | $\sim$ | $\sim$ | $\sim$ | $1.104 \pm 0.037$ |

Table 3: Offline and Online RL Algorithm Performance Across Multiple Offline Datasets. Each of the first three column headers indicates the baseline algorithm that collected the offline dataset where "thrpt_argmax" is an alias for throughput_argmax. Each row header (except "baseline", "PPO", and "SAC") is an offline RL algorithm trained on one of three offline datasets. "baseline" refers to the performance of the original baseline heuristic policies without any offline data collection. "(norm)" indicates that the algorithm implements state-normalization based on the offline dataset while "(no norm)" indicates that the algorithm does not. If not specified, the algorithm does not implement state normalization. We use $\alpha = 1.0$ and $\beta = 10.0$ in our evaluation of PTD3.

| $\beta$ | dataset-generating algorithm | | |
|---|---|---|---|
| | throughput_argmax | system_default | utility_logistic |
| 0.1 | $0.744 \pm 0.056$ | $0.878 \pm 0.048$ | $0.816 \pm 0.045$ |
| 0.3 | $0.744 \pm 0.056$ | $\mathbf{0.958 \pm 0.041}$ | $1.044 \pm 0.029$ |
| 1.0 | $0.744 \pm 0.056$ | $\mathbf{0.974 \pm 0.040}$ | $1.015 \pm 0.045$ |
| 3.0 | $0.744 \pm 0.056$ | $\mathbf{0.995 \pm 0.044}$ | $1.022 \pm 0.041$ |
| 10.0 | $0.744 \pm 0.057$ | $\mathbf{1.017 \pm 0.044}$ | $1.083 \pm 0.047$ |
| 30.0 | $0.744 \pm 0.056$ | $0.679 \pm 0.044$ | $\mathbf{1.209 \pm 0.045}$ |
| 100.0 | $0.744 \pm 0.056$ | $0.213 \pm 0.070$ | $\mathbf{1.226 \pm 0.049}$ |
| 300.0 | $0.744 \pm 0.056$ | $0.243 \pm 0.059$ | $\mathbf{1.252 \pm 0.063}$ |

Table 4: PTD3 Performance where $\alpha = 1.0$. For each of the algorithms, we evaluate its performance on 32 evaluation episodes, each of which is 3200 steps. We avoid bolding the throughput_argmax runs, as they all have roughly the same performance.

findings strongly suggest that it would be imprudent to deploy such algorithms trained on a similar task into the real world, even if they were trained on datasets collected from *real* interactions.

Since our implementation of PTD3 is, on average, able to significantly outperform not only the heuristic baseline policies, but also several existing state-of-the-art offline RL algorithms, this suggests that the poor performance across existing algorithms is not due to a lack of coverage across datasets, but rather the lack of diversity and breadth of testing environments for these algorithms. While the D4RL benchmark has become a standard for assessing offline RL performance, it is essential to recognize its limitations. Algorithms that are touted as state-of-the-art based on their performance on D4RL may not generalize well to other, perhaps more complex or varied, scenarios. We have shown that many advanced offline RL algorithms have the potential to fail catastrophically when deployed in different contexts or faced with unfamiliar environments. Therefore, to ensure robustness and reliability, it is crucial to test offline RL algorithms across a wider array of datasets and environments. This broader testing approach helps to uncover potential weaknesses and provides a more comprehensive understanding of an algorithm's capabilities and limitations. We hope that by expanding the scope of testing beyond popular benchmarks like D4RL and RL Unplugged,

| $\beta$ | dataset-generating algorithm | | |
|---|---|---|---|
| | throughput_default | delay_default | utility_default |
| 0.1 | $0.744 \pm 0.056$ | $0.878 \pm 0.048$ | $0.816 \pm 0.045$ |
| 0.3 | $0.744 \pm 0.056$ | $\mathbf{0.958 \pm 0.041}$ | $1.044 \pm 0.029$ |
| 1.0 | $0.744 \pm 0.056$ | $\mathbf{0.974 \pm 0.040}$ | $1.015 \pm 0.045$ |
| 3.0 | $0.744 \pm 0.056$ | $\mathbf{0.995 \pm 0.044}$ | $1.022 \pm 0.041$ |
| 10.0 | $0.744 \pm 0.057$ | $\mathbf{1.017 \pm 0.044}$ | $1.083 \pm 0.047$ |
| 30.0 | $0.744 \pm 0.056$ | $0.679 \pm 0.044$ | $\mathbf{1.209 \pm 0.045}$ |
| 100.0 | $0.744 \pm 0.056$ | $0.213 \pm 0.070$ | $\mathbf{1.226 \pm 0.049}$ |
| 300.0 | $0.744 \pm 0.056$ | $0.243 \pm 0.059$ | $\mathbf{1.252 \pm 0.063}$ |

Table 5: PTD3 Performance where $\alpha = 1.0$. For each of the algorithms, we evaluate its performance on 32 evaluation episodes, each of which is 3200 steps. We avoid bolding the throughput_argmax runs, as they all have roughly the same performance.

| Offline Dataset | BC (no norm) | PTD3 ($\beta =$ "$\infty$") |
|---|---|---|
| **throughput_argmax** | $0.751 \pm 0.055$ | $0.770 \pm 0.055$ |
| **system_default** | $0.432 \pm 0.063$ | $0.547 \pm 0.059$ |
| **utility_logistic** | $0.948 \pm 0.042$ | $\mathbf{1.252 \pm 0.079}$ |

Table 6: Comparison between Behavioral Cloning and $Q$-function pessimism. We evaluate each offline RL algorithm across 32 evaluation episodes, each of which is 3200 steps. In this implementation of PTD3, we set $\alpha = 1.0$, set $\beta = 1.0$, and manually remove the $Q$-value maximization term from the policy-update step to simulate $\beta = \infty$.

researchers and practitioners can better gauge the true potential and practicality of their offline RL solutions.

**Behavioral Cloning vs. State-Action Value Function Pessimism.** In testing PTD3 ($\alpha = 1.0$) on all three offline datasets with varying values of $\beta$, we note that while the performance on the system_default dataset improves up to a point as $\beta$ increases then drops off, the performance on the utility_logistic dataset improves substantially even up to values as high as $\beta = 300.0$. In fact, for large enough $\beta$, the performance of PTD3 on the utility_logistic dataset is comparable to that of the best performing *online* deep RL algorithm, PPO. This behavior leads us to question whether or not $Q$-function pessimism as we've defined it in this work is always comparable to behavioral cloning. To further test this, we compare the performance of behavioral cloning (without offline dataset feature normalization) with PTD3 where the $Q$-value maximization term is removed from the policy-update step. In other words, this implementation of PTD3 would be performing pure minimization of the $Q$-value uncertainty estimates with respect to state-action pairs, without any regard for how high those $Q$-values are. The results are illustrated in Table 6. Interestingly, we note that the performance of this purely pessimistic PTD3 implementation is significantly higher than that of behavioral cloning when both are trained on the utility_logistic offline dataset, while the two implementations are comparable in the case of the other two datasets. This is reflective of a fundamental difference between behavioral cloning and $Q$-value uncertainty minimzation: while the objective of a behavioral cloning agent is to pointwise match the agent's actions to those chosen by the behavior policy, the objective of the uncertainty-minimizing agent is to choose actions that minimize the uncertainty in the $Q$-function estimates by considering the associated variance in $Q$-values.

**Limitations.** Several limitations exist in our current approach. First, the NetworkGym environment simulation setup that we use in all experiments assumes a fixed number of UEs. Consequently, the addition or removal of even a single UE necessitates retraining the online or offline algorithms from scratch, given the current formulation of the MDP. Additionally, our simulation setting incorporates unrealistic degenerate movement patterns for UEs, which may not accurately reflect real-world dynamics. Finally, one of the major limitations of PTD3 is the constraint on the parameter size of the $Q$-networks: if the $Q$-networks each have $d$ parameters, then $d \times d$ space is required to store $\widetilde{\mathbf{F}}_t$ in memory. As a result, in order for gradient-related computations to fit on our 12 GB NVIDIA

TITAN Xp, we were required to use MLP critics with two hidden layers of 64 neurons each instead of hidden layers of 400 and 300 neurons as TD3+BC uses.

**Future Work.** In future work, we plan to explore methods to appropriately featurize multiple UEs to allow for dynamic changes in their number without requiring retraining any algorithms. This will involve rethinking the current MDP formulation to accommodate a variable number of UEs more flexibly. We also aim to incorporate more complex movement patterns for UEs, such as random walks, to gain a better understanding of how our tested algorithms generalize to these settings. In addition to the previously mentioned areas, potential opportunities exist to enhance the performance of our PTD3 algorithm. The use of GPUs with larger memory capacities would enable us to use larger critic network architectures for PTD3. Additionally, in this work, we primarily explore estimating $\mathbf{F}_t$ using an exponentially-weighted moving sum with low variance ($\alpha = 1.0$); this is because as $\beta$ changes, training with a high variance estimator of the Fisher information matrix across timesteps makes it difficult to properly evaluate the effect of $\beta$.

## 6    Conclusion

In this work, we present NetworkGym, a high-fidelity gym-like network environment simulator that facilitates multi-access traffic splitting. NetworkGym seamlessly aids in training and evaluating online and offline RL algorithms on the multi-access traffic splitting task in simulation. In our simulated experiments, we demonstrate that existing state-of-the-art offline RL algorithms fail to significantly outperform heuristic policies on this task. This highlights the critical need for a broader range of benchmarks across multiple domains for offline RL algorithm evaluation. On the other hand, our proposed PTD3 algorithm significantly outperforms not only heuristic policies, but also many state-of-the-art offline RL algorithms trained on heuristic-generated datasets. These findings pave the way for more effective offline RL algorithms and demonstrate the potential of PTD3 as a strong contender among existing solutions. Future research should consider evaluating offline RL algorithms on networking-specific tasks alongside other benchmarks to foster the development of more robust and versatile solutions.

## Acknowledgments and Disclosure of Funding

The work is partially supported by National Science Foundation (NSF) Award #2007117 and Award #2003257 under NSF/Intel Partnership on Machine Learning for Wireless Networking Systems (ML-WiNS).

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

# Appendix

## A    Technical Details Concerning GMA Protocol

Both the GMA and ATSSS protocols provide mechanisms for flexible selection of network paths and leverage network intelligence and policies to dynamically adapt traffic distribution across selected paths under changing network/link conditions. Generally, a multi-access network protocol, e.g. GMA, consists of the following two sublayers:

- Convergence sublayer: This layer performs multi-access specific tasks such as access (path) selection, multi-link (path) aggregation, splitting/reordering, lossless switching, keep-alive, and probing Kanugovi et al. [2020].

- Adaptation sublayer: This layer performs functions to handle tunneling, network layer security, and network address translation (NAT). This design only operates at the physical and routing layers in the network data plane and does not require any modifications to the higher layer protocols, such as user datagram protocol (UDP), transport control protocol (TCP), IP security (IPSec), etc.

On the other hand, maximizing the benefits of a multi-access system necessitates solving a decision problem—intelligently distributing user data traffic across available access links to optimize user experience while making the best use of available radio resources. To effectively manage traffic in a multi-access network, it's crucial to incorporate measurements that reflect the varying connectivity conditions of different networks. For instance, end-to-end (e2e) packet delay measurements can help determine which access network offers better latency performance. Similarly, for quality-of-service (QoS) flows that demand high reliability, the packet drop ratio can indicate the necessity for redundant transmission across multiple networks. Besides e2e packet statistics, Radio Access Network (RAN) measurements, like reference signal received power (RSRP), reference signal received quality (RSRQ), and received signal strength indicator (RSSI), can reveal any degradation in network quality due to issues like deteriorating radio link quality or congestion in real-time. However, integrating these diverse data sources—including e2e packet statistics and RAN measurements—to formulate an optimal traffic management algorithm for multi-access networks remains a complex challenge.

## B    Offline RL Using Online Algorithms

Many recent state-of-the-art offline RL methods make minor modifications to existing online RL algorithms. For example, SAC-N modifies a popular online RL algorithm known as Soft Actor-Critic (SAC) by simply increasing the number of $Q$-function approximators from 2 to $N > 2$ in order to further mitigate the over-estimation of $Q$-values Haarnoja et al. [2018a], An et al. [2021]. Ensemble-Diversified Actor-Critic (EDAC) adds a regularization term that minimizes the pairwise cosine similarity of the gradients across different $Q$-function approximators to incentivize the policy network to choose actions at which the gradients of the $Q$-function networks have high alignment An et al. [2021]. Nikulin et al. Nikulin et al. [2022] propose Large Batch SAC (LB-SAC), which avoids the need to use a large ensemble of $Q$-function networks and instead scales the batch size used to train these networks with the result of improving learning duration while maintaining performance.

## C    Pessimistic TD3

In this section, we introduce a new offline RL algorithm, which we denote as Pessimistic TD3 (PTD3). We highlight that the TD3+BC algorithm from Fujimoto et al. removes online access to the environment and instead replaces the replay buffer $\mathcal{B}$ with an offline dataset $\mathcal{D}$ Fujimoto and Gu [2021]. Additionally, TD3+BC adds a simple behavioral cloning term to the deterministic policy gradient step with the goal of minimizing the square-difference between the actions in the dataset and the corresponding actions output by the learned policy Fujimoto and Gu [2021]. In order to instead incorporate pessimism into the TD3 algorithm to produce PTD3, we first compute a Fisher information matrix of the offline dataset on one of the critics at each policy update step as in Equation

2.

$$\mathbf{F}_t = \sum_{k=1}^{K} \nabla Q_{\theta_1(t)}(s_k, a_k) \nabla^{\mathrm{T}} Q_{\theta_1(t)}(s_k, a_k) + \lambda_r \mathbf{I}_d \tag{2}$$

where $\lambda_r$ is a hyperparameter and $\mathbf{I}_d$ is the $d \times d$ identity matrix to ensure that $\mathbf{F}_t$ is invertible. The gradient of the state-action value function is represented as a $d$-dimensional column vector where the $Q$-network parameter vector $\theta_i \in \mathbb{R}^d$. From there, we can use Equation 3 to compute a statistically motivated estimate of the uncertainty in the $Q$-values at state-action pairs where states are sampled from the dataset and actions are chosen from the learned policy.

$$\Gamma_t = \mathbb{E}_{s \sim \mathcal{D}} \left[ \beta \sqrt{\nabla^{\mathrm{T}} Q_{\theta_1(t)}(s, \pi_\phi(s)) \mathbf{F}_t^{-1} \nabla Q_{\theta_1(t)}(s, \pi_\phi(s))} \right] \tag{3}$$

This allows us to compute a pessimistic estimate of the $Q$-values associated with the states sampled from the dataset and actions taken from the learned policy as in Equation 4.

$$\bar{Q}_t = \mathbb{E}_{s \sim \mathcal{D}} \left[ Q_{\theta_1(t)}(s, \pi_\phi(s)) \right] - \Gamma_t \tag{4}$$

Finally, we can use the deterministic policy gradient to update the policy parameters in the gradient direction that maximizes this lower-bound estimate $\bar{Q}_t$.

In practice, computing $\mathbf{F}_t$ across the entire dataset on each iteration is quite computationally expensive, as it requires $K = |\mathcal{D}|$ per-sample-gradient calculations. One way we work around this issue is by sampling a large batch (we use $2^{14}$ samples) from the dataset and estimating $\mathbf{F}_t$ over that batch instead. Additionally, computing the inverse of $\mathbf{F}_t$ has a time complexity of roughly $O(d^3)$, which constrains the dimension of the $Q$-networks from being too large. In order to better combat these issues in practice, we instead initialize the estimator $\widetilde{\mathbf{F}}_0 = \mathbf{I}_d$ and update $\widetilde{\mathbf{F}}_t$ as an exponentially-weighted moving sum over previous iterations via Equation 5.

$$\widetilde{\mathbf{F}}_t = \alpha \widetilde{\mathbf{F}}_{t-1} + \nabla Q_{\theta_1(t)}(s_i, a_i) \nabla^{\mathrm{T}} Q_{\theta_1(t)}(s_i, a_i) \tag{5}$$

where $\alpha \in (0, 1]$ is a parameter that controls the bias-variance trade-off in the $\widetilde{\mathbf{F}}_t$ estimator and $(s_i, a_i) \sim \mathcal{D}$ is a single tuple sampled from the dataset. In this way, $\mathbf{F}_t$ is analogous to a "full-batch gradient descent" calculation while $\widetilde{\mathbf{F}}_t$ is analogous to a "stochastic gradient descent." As $\alpha \to 0$, the resulting estimator has higher variance, but is less biased by previous values of $\widetilde{\mathbf{F}}_t$. On the other hand, as $\alpha \to 1$, the variance of the $\widetilde{\mathbf{F}}_t$ estimator reduces, but the estimator retains information with respect to older $Q$-network parameters, which are more likely to be obsolete.

As a result of estimating $\mathbf{F}_t$ in this way, we may use the Sherman-Morrison formula to compute its inverse on each iteration and avoid the $O(d^3)$ complexity required to recompute the inverse from scratch. This reduces the runtime of the algorithm by roughly a factor of 3:

$$\widetilde{\mathbf{F}}_t^{-1} = \frac{1}{\alpha} \widetilde{\mathbf{F}}_{t-1}^{-1} - \frac{\widetilde{\mathbf{F}}_{t-1}^{-1} \nabla Q_{\theta_1(t)}(s_i, a_i) \nabla^{\mathrm{T}} Q_{\theta_1(t)}(s_i, a_i) \widetilde{\mathbf{F}}_{t-1}^{-1}}{\alpha^2 + \alpha \nabla^{\mathrm{T}} Q_{\theta_1(t)}(s_i, a_i) \widetilde{\mathbf{F}}_{t-1}^{-1} \nabla Q_{\theta_1(t)}(s_i, a_i)} \tag{6}$$

In practice, we add a small amount of Gaussian noise $n \sim \mathcal{N}(\mathbf{0}, \epsilon \mathbf{I}_d)$ to the gradient vectors, before incorporating them into the $\widetilde{\mathbf{F}}_t$ estimator.[3] This ensures that $\widetilde{\mathbf{F}}_t$ remains invertible over a large number of iterations if $\alpha$ is not close to 1. Additionally, due to cumulative numerical round-off error from repeatedly applying the rank-1 update rule, we recompute $\widetilde{\mathbf{F}}_t^{-1}$ from scratch every 100 iterations to prevent it from diverging too far from its true value. We present the final version of PTD3 in Algorithm 1 with the relevant modifications to TD3+BC highlighted.

---

[3]In our experiments, we use $\epsilon = 10^{-9}$.

**Algorithm 1** Pessimistic TD3 (PTD3) with Biased $\mathbf{F}_t$ Estimator

1: Initialize critic networks $Q_{\theta_1}$, $Q_{\theta_2}$, and actor network $\pi_\phi$ with random parameters $\theta_1, \theta_2, \phi$
2: Initialize target networks $\theta'_1 \leftarrow \theta_1$, $\theta'_2 \leftarrow \theta_2$, $\phi' \leftarrow \phi$
3: Initialize Fisher information matrix estimator $\widetilde{\mathbf{F}}_0 \leftarrow \mathbf{I}_d$
4: Initialize offline dataset $\mathcal{D} = \{(s_k, a_k, r_k, s'_k)\}_{k=1}^K$
5: **for** $t = 1$ to $T$ **do**
6: $\quad$ Sample mini-batch of $N$ transitions $(s, a, r, s')$ from $\mathcal{D}$
7: $\quad$ $\tilde{a} \leftarrow \pi_{\phi'}(s') + \epsilon$, $\epsilon \sim \text{clip}(\mathcal{N}(0, \tilde{\sigma}), -c, c)$
8: $\quad$ $y \leftarrow r + \gamma \min_{i=1,2} Q_{\theta'_i}(s', \tilde{a})$
9: $\quad$ Update critics $\theta_i \leftarrow \arg\min_{\theta_i} N^{-1} \sum (y - Q_{\theta_i}(s, a))^2$
10: $\quad$ **if** $t$ mod $d$ **then**
11: $\quad\quad$ Sample single transition $(s_j, a_j, r_j, s'_j)$ from $\mathcal{D}$
12: $\quad\quad$ $\widetilde{\mathbf{F}}_t \leftarrow \alpha \widetilde{\mathbf{F}}_{t-1} + \nabla Q_{\theta_1(t)}(s_j, a_j) \nabla^{\mathrm{T}} Q_{\theta_1(t)}(s_j, a_j)$
13: $\quad\quad$ Compute $\widetilde{\mathbf{F}}_t^{-1}$ using rank-1 update rule (6)
14: $\quad\quad$ Update $\phi$ by the deterministic policy gradient:
15: $\quad\quad$ $\nabla_\phi J(\phi) = N^{-1} \sum \nabla_\phi \left[ Q_{\theta_1}(s, \pi_\phi(s)) - \beta \sqrt{\nabla_{\theta_1}^{\mathrm{T}} Q_{\theta_1}(s, \pi_\phi(s)) \widetilde{\mathbf{F}}_t^{-1} \nabla_{\theta_1} Q_{\theta_1}(s, \pi_\phi(s))} \right]$
16: $\quad\quad$ Update target networks:
17: $\quad\quad$ $\theta'_i \leftarrow \tau \theta_i + (1 - \tau)\theta'_i$
18: $\quad\quad$ $\phi' \leftarrow \tau\phi + (1 - \tau)\phi'$
19: $\quad$ **end if**
20: **end for**

## D NetworkGym Environment Configuration

We reproduce the relevant NetworkGym `env_config` associated with our experiments in Listing 1. We place asterisks (*) at the `steps_per_episode` and `random_seed` parameters, because we modify these values across different experiments to support different purposes.

Listing 1: Environment configuration for NetworkGym.

```
{
    "type": "env-start",
    "subscribed_network_stats": [ // environment will only
        ↪ report subscribed measurements.
        "wifi::dl::max_rate",
        "wifi::ul::max_rate",
        "wifi::cell_id",
        "lte::dl::max_rate",
        "lte::cell_id",
        "lte::slice_id",
        "lte::dl::rb_usage",
        "lte::dl::cell::max_rate",
        "lte::dl::cell::rb_usage",
        "gma::x_loc",
        "gma::y_loc",
        "gma::dl::missed_action",
        "gma::dl::measurement_ok",
        "gma::dl::tx_rate",
        "gma::dl::delay_violation",
        "gma::dl::delay_test_1_violation",
        "gma::dl::delay_test_2_violation",
        "gma::dl::rate",
        "gma::wifi::dl::rate",
        "gma::lte::dl::rate",
        "gma::dl::qos_rate",
        "gma::wifi::dl::qos_rate",
```

```
        "gma::lte::dl::qos_rate",
        "gma::dl::owd",
        "gma::wifi::dl::owd",
        "gma::lte::dl::owd",
        "gma::dl::max_owd",
        "gma::wifi::dl::max_owd",
        "gma::lte::dl::max_owd",
        "gma::wifi::dl::priority",
        "gma::lte::dl::priority",
        "gma::wifi::dl::traffic_ratio",
        "gma::lte::dl::traffic_ratio",
        "gma::wifi::dl::split_ratio",
        "gma::lte::dl::split_ratio",
        "gma::ul::missed_action",
        "gma::ul::measurement_ok",
        "gma::ul::tx_rate",
        "gma::ul::delay_violation",
        "gma::ul::delay_test_1_violation",
        "gma::ul::delay_test_2_violation",
        "gma::ul::rate",
        "gma::wifi::ul::rate",
        "gma::lte::ul::rate",
        "gma::ul::qos_rate",
        "gma::wifi::ul::qos_rate",
        "gma::lte::ul::qos_rate",
        "gma::ul::owd",
        "gma::wifi::ul::owd",
        "gma::lte::ul::owd",
        "gma::ul::max_owd",
        "gma::wifi::ul::max_owd",
        "gma::lte::ul::max_owd",
        "gma::wifi::ul::priority",
        "gma::lte::ul::priority",
        "gma::wifi::ul::traffic_ratio",
        "gma::lte::ul::traffic_ratio",
        "gma::wifi::ul::split_ratio",
        "gma::lte::ul::split_ratio"
    ],
    "steps_per_episode": *, // number of steps per each episode
        ↪   - last step is given a truncated signal.
    "episodes_per_session": 1, // always set to 1 - every
        ↪ episode is treated in the infinite-horizon setting.
    "random_seed": *, // random seed ONLY affects UE placement
        ↪ and movement in environment.
    "downlink_traffic": true, // simulates downlink data flow
    "max_wait_time_for_action_ms": -1, // max time network gym
        ↪ worker will wait for an action (capped to 10 minutes)
        ↪ .
    "enb_locations": { // x, y and z locations of singular base
        ↪   station.
        "x": 40,
        "y": 0,
        "z": 3
    },
    "ap_locations": [ // x, y and z locations of Wi-Fi access
        ↪ point(s).
        {
            "x": 30,
            "y": 0,
```

```
                "z": 3
            },
            {
                "x": 50,
                "y": 0,
                "z": 3
            }
    ],
    "num_users": 4,
    "user_random_walk": { // in this model, each user begins
        ↪ moving to the right at 1 m/s and bounces back and
        ↪ forth for the entire duration.
        "min_speed_m/s": 1,
        "max_speed_m/s": 1,
        "min_direction_gradients": 0.0,
        "max_direction_gradients": 0.0,
        "distance_m": 1000000
    },
    "user_location_range": { // initially, users will be
        ↪ randomly deployed within this x, y range depending on
        ↪  random_seed.
        "min_x": 0,
        "max_x": 80,
        "min_y": 0,
        "max_y": 0,
        "z": 1.5
    },
    "measurement_start_time_ms": 1000, // the first measurement
        ↪  start time. The first measurement will be sent to
        ↪ the agent between [measurement_start_time_ms,
        ↪ measurement_start_time_ms + measurement_interval_ms].
    "transport_protocol": "tcp",
    "udp_poisson_arrival": true, // only used for UDP.
    "min_udp_rate_per_user_mbps": 6, // only used for UDP.
    "max_udp_rate_per_user_mbps": 6, // only used for UDP.
    "qos_requirement": { // only used for qos_steer environment
        ↪ .
        "delay_bound_ms": 1000,
        "delay_test_1_thresh_ms": 2000,
        "delay_test_2_thresh_ms": 4000
    },
    "GMA": {
        "downlink_mode": "split",
        "uplink_mode": "auto", // under "auto", TCP ACK will "
            ↪ steer" and TCP data will "split".
        "enable_dynamic_flow_prioritization": false, // because
            ↪  we do not use UDP traffic with QoS requirement,
            ↪ we don't use DFP.
        "measurement_interval_ms": 100, // duration of a
            ↪ measurement interval.
        "measurement_guard_interval_ms": 0 // no gap between
            ↪ measurement intervals.
    },
    "Wi-Fi": {
        "ap_share_same_band": false, // APs use different
            ↪ frequency bands.
        "enable_rx_signal_based_handover": true, // always
            ↪ connect to Wi-Fi AP with strongest Rx signal (the
            ↪  Rx signal is measured from BEACONS).
```

```
        "measurement_interval_ms": 100,
        "measurement_guard_interval_ms": 0
    },
    "LTE": {
        "resource_block_num": 50, // number of resouce blocks
            ↪ for LTE: 25 for 5 MHZ, 50 for 10 MHZ, 75 for 15
            ↪ MHZ and 100 for 20 MHZ.
        "measurement_interval_ms": 100,
        "measurement_guard_interval_ms": 0
    }
}
```

We open-source our primary code and offline datasets at `github.com/hmomin/networkgym`. Each section (except Section G) in this document references assets relative to the root directory of this repository.

## E   Computational Resources

We make use of four internal 12 GB NVIDIA TITAN Xp GPUs to perform our experiments. With these GPUS, to perform all experiments described in this document requires roughly 1 month of compute, assuming each of 8 different CPU processes is used to perform an agent evaluation. Using only a single process to perform agent evaluation would result in the compute increasing to roughly 3 months.

## F   Offline Data Collection

For each of three different heuristic policies (`throughput_argmax`, `system_default`, and `utility_logistic`), we collect and store 64 episodes of offline data on our Network-Gym Multi-Access Traffic Splitting environment (denoted `nqos_split`). Each episode contains 10,000 steps worth of data. The associated configuration file (located at `network_gym_client/envs/nqos_split/config.json`) for the episodes is chosen with the following constraints in mind:

- At initialization of each environment, four UEs are randomly stationed 1.5 meters above the $x$-axis between $x = 0$ and $x = 80$ meters. From there, they begin to bounce back and forth in the $x$-direction at 1 m/s for the entire duration of an episode.

- The Wi-Fi access points are stationed at $(x, z) = (30\text{m}, 3\text{m})$ and $(x, z) = (50\text{m}, 3\text{m})$, respectively.

- The LTE base station lies at $(x, z) = (40\text{m}, 3\text{m})$.

- The only change in the configuration file between episodes is the `random_seed` parameter. We use random seed values from 0 to 63, inclusive, for this parameter.

We store the resulting three offline datasets in the `NetworkAgent/buffers` directory. Each dataset is a folder that contains 64 `.pickle` files, one for each episode. Each `.pickle` file contains a tuple of four `numpy` arrays in the following order: (states, actions, rewards, next states) with shapes ([9999, 56], [9999, 4], [9999, 1], [9999, 56]), respectively.

We also provide a shell script (`offline_collection.sh`) to generate data for offline learning. The heuristic policy that takes actions in the environments can be specified at the top of the script.

## G   Training Existing State-of-the-Art Offline RL Algorithms

To test several existing state-of-the-art offline reinforcement learning (RL) algorithms, we make use of the Clean Offline RL library provided at `github.com/tinkoff-ai/CORL`, which uses the Apache 2.0 license. More specifically, we modify their library at `github.com/hmomin/CORL-compare` to be compatible with our offline dataset generated on the

NetworkGym simulator. The modifications we make to the offline RL algorithm files (located at `algorithms/offline`) only support the following purposes:

- We switch the algorithmic implementations from using D4RL-specific loading to using our NetworkGym `OfflineEnv` class instead.
- We remove all resulting unused D4RL-specific environment/dataset loading and evaluation code.
- We modify the `env` parameter in the `TrainConfig` class for each algorithm to use an environment specified by one of our three offline datasets.
- We modify the `normalize` boolean parameter (where applicable) in the `TrainConfig` class to toggle whether or not we would like the algorithm to perform feature normalization based on the offline dataset.

Using these modifications, any of the algorithm scripts at `algorithms/offline` can be executed directly to train these algorithms. We use the default hyperparameters for all algorithms, except where we toggle the `normalize` parameter.

## H   Training PTD3

To train our implementation of Pessimistic TD3 (PTD3), we use the default hyperparameters in TD3+BC, except for the following modifications:

- We train PTD3 for 10,000 steps, instead of 1,000,000 steps, which we do for TD3+BC.
- We test PTD3 across various values of $\alpha$ and $\beta$; we then report the corresponding experimental results.

We provide the shell script `train_offline_ptd3.sh` to train PTD3 on any offline dataset generated by one of our heuristic algorithms. The desired values of offline dataset, $\alpha$, and $\beta$ can be specified at the top of the script.

## I   Training Online Deep RL Algorithms

We use `stable-baselines3` to train two different online deep RL algorithms, PPO and SAC. We do so by initializing a random agent, then updating that agent through 8 successive phases. In each phase, we parallelize environment instantiations across 8 different random seeds, where each environment runs for 10,000 steps, resulting in a total of 64 different environment instantiations. In this way, the online learning algorithm trains across the same number of steps available in each of the offline datasets, to allow for proper comparison. Additionally, for our parallel environment random seeds, we use 0-7, inclusive, followed by 8-15, 16-23, ..., 56-63. We provide the shell script, `train_online_parallel.sh`, in order to perform this training process with PPO and SAC. We use the default hyperparameters specified by `stable-baselines3`.

## J   Evaluating Trained Agents

Finally, to evaluate a trained agent (whether online or offline), we place the resulting model file in the `NetworkAgent/models` directory. Then, the model filename (without extension) can be specified as the `agent` parameter at the top of the `test_agent.sh` shell script and the script can be executed to evaluate the agent on a single 3,200 step episode. In our experiments, we evaluate each agent across 32 or 40 episodes (each with a different `random_seed` parameter), depending on the experiment. Each episode is 3,200 steps and the `random_seed` parameter takes on values between 128-159, inclusive, for 32 evaluation episodes or 128-167, inclusive, for 40 evaluation episodes. We otherwise use the same environment configuration details mentioned in Section Offline Data Collection.

