# OpenReview forum: "NetworkGym: Reinforcement Learning Environments for Multi-Access Traffic Management in Network Simulation"
_NeurIPS.cc/2024/Datasets_and_Benchmarks_Track — NeurIPS 2024 Track Datasets and Benchmarks Poster_

### Official Review · Reviewer_jRFS · 2024-07-11
**Review of paper 2560**

**Rating:** 8
**Confidence:** 5
**Clarity:** The paper is clearly well written and…

**Review:**

The paper is well written, and the contributions are clearly stated. Related work is also clearly presented. The originality of the paper is thus easy to understand.

**Strengths:**

The problem to tackle is quite an important one, especially considering the evolution of the telecommunication standards, which the authors have clearly understood. The authors have developed an environment that clearly explains how to tackle this issue using RL.

**Additional Feedback:**

NA

**Correctness:**

The claims are correct. Some improvement on the evaluation method have been suggested above.

**Documentation:**

Yes.

**Ethics:**

No.

**Limitations:**

The authors have clearly stated the limitations of their work. Some addition are suggested above.

**Opportunities For Improvement:**

Some details would benefit the quality of the paper:
- Addition of a reference for behavioral cloning
- In Section 3, when describing the Markov Decision processes, it is not clear why éS & A can both be potentially infinite OR continuous". Why using "or"?
- In Section 3, I believe a better explanation of the action space would help, especially the splitting ratio. Why limiting it to Wi-fi?
-The link between the achieved results (Section 4) and the problem setup (Section 3) is not clear. Could you explicit it in more details, especially the link with the eigenvalues?
- In the limitations, I would add the fact that, in practice, a varying number of channels/communication means is quite normal.

**Relation To Prior Work:**

Yes, the related work is clearly stated.

**Summary And Contributions:**

The paper presents a clear environment to evaluate RL-based approaches to study multi-access traffic management solutions.

---

### Official Review · Reviewer_pxPu · 2024-07-24
**NetworkGym: Reinforcement Learning Environments for Multi-Access Traffic Management in Network Simulation**

**Rating:** 6
**Confidence:** 4
**Correctness:** Yes
**Clarity:** Yes

**Review:**

The paper introduces NetworkGym, a high-fidelity network simulation environment designed to facilitate the training and evaluation of reinforcement learning (RL) algorithms for multi-access traffic management. The platform supports both online and offline RL, providing a comprehensive tool for researchers to develop and benchmark RL-based solutions for network optimization.

NetworkGym stands out for its comprehensive simulation environment, which is realistic and detailed, supporting multiple programming languages and making it accessible and flexible for various use cases. The environment supports a wide range of RL algorithms and includes pre-specified benchmarks, allowing for thorough evaluation and comparison. Customizable reward functions further enhance its utility. The paper provides extensive empirical validation, evaluating 14 RL algorithms and highlighting the strengths and weaknesses of current methods. The proposed PTD3 algorithm shows significant improvements over existing state-of-the-art offline RL algorithms. The focus on real-world relevance is another strength, as NetworkGym addresses the critical challenge of multi-access traffic splitting by simulating dynamic and realistic network scenarios.

However, there are several weaknesses. The simulation setup assumes a fixed number of UEs, requiring retraining if the number changes, which reduces flexibility. Additionally, the environment incorporates unrealistic movement patterns for UEs, which may limit the applicability of the results. The PTD3 algorithm's memory constraints necessitate smaller network architectures, potentially limiting performance. The evaluation of several state-of-the-art RL algorithms shows erratic performance and poor generalization across different datasets, indicating the need for more robust evaluation methods. The computation of the Fisher information matrix (Ft) for PTD3 is computationally intensive, requiring significant resources. The detailed implementation of PTD3 adds complexity, which might deter adoption.

To improve, the authors could develop in future methods to handle dynamic changes in the number of UEs without requiring retraining and incorporate more realistic UE movement patterns to enhance the realism of simulations. Exploring scalable solutions for handling larger Q-networks without exceeding memory limitations would also be beneficial. Conducting broader testing across a wider range of datasets and environments could help uncover potential weaknesses and better understand the algorithms' capabilities.

Addressing these areas would enhance the robustness, flexibility, and applicability of NetworkGym and the PTD3 algorithm, making them more valuable tools for the research community.

**Strengths:**

1. Comprehensive Simulation Environment:
        ◦ High-Fidelity Simulation: NetworkGym provides a high-fidelity network environment simulator designed specifically for multi-access traffic management, allowing for realistic and detailed simulations.
        ◦ Open-Source and Accessible: The environment is open-source, making it accessible for researchers and practitioners to use, modify, and extend.
        ◦ Support for Multiple Programming Languages: The separation of Agent and Environment allows the use of different programming languages for ML algorithms and network simulations, enhancing flexibility and usability.
2. Facilitation of RL Research:
        ◦ Support for Both Online and Offline RL: NetworkGym supports both online and offline reinforcement learning (RL), providing a versatile platform for testing and evaluation.
        ◦ Pre-Specified Benchmarks: The environment includes pre-specified natural language benchmark tasks and supports popular RL algorithms, facilitating comprehensive evaluation and benchmarking.
        ◦ Customizable Reward Functions: Users can specify different reward functions to meet various Quality of Service (QoS) requirements, enhancing the utility of the environment for diverse applications.
3. Empirical Validation:
        ◦ Extensive Evaluation: The paper provides an extensive evaluation of 14 popular RL algorithms, highlighting the strengths and weaknesses of current state-of-the-art methods in multi-access traffic management.
        ◦ Introduction of PTD3 Algorithm: The proposed Pessimistic TD3 (PTD3) algorithm shows significant improvement over existing state-of-the-art offline RL algorithms, demonstrating the potential of incorporating value-function pessimism in RL.
4. Addressing Real-World Challenges:
        ◦ Focus on Traffic Splitting: The environment is tailored to address the specific challenge of multi-access traffic splitting, a critical aspect of modern and future network management.
        ◦ Dynamic and Realistic Scenarios: NetworkGym allows for the simulation of dynamic and realistic network scenarios, including varying user equipment (UE) movement patterns and heterogeneous network conditions.

**Additional Feedback:**

No additional feedback.

**Documentation:**

NA

**Ethics:**

No ethic issues.

**Limitations:**

1. Dynamic UE Handling:
        ◦ Develop methods to handle dynamic changes in the number of UEs without requiring algorithm retraining. This could involve rethinking the current MDP formulation to accommodate a variable number of UEs more flexibly.
2. Realistic Movement Patterns:
        ◦ Incorporate more complex and realistic movement patterns for UEs, such as random walks or mobility models based on real-world data. This would enhance the realism and applicability of the simulation results.
3. Scalable Memory Solutions:
        ◦ Explore scalable solutions to handle larger Q-networks without exceeding memory limitations. This could involve using GPUs with larger memory capacities or optimizing the memory usage of the algorithm.
4. Broader Testing and Evaluation:
        ◦ Conduct broader testing and evaluation of RL algorithms across a wider range of datasets and environments. This would help uncover potential weaknesses and provide a more comprehensive understanding of the algorithms' capabilities and limitations.

**Opportunities For Improvement:**

1. Fixed Number of UEs:
        ◦ The current simulation setup assumes a fixed number of UEs, requiring retraining of RL algorithms if the number of UEs changes. This limitation reduces the flexibility and generalizability of the simulation environment.
2. Simplistic Movement Patterns:
        ◦ The environment currently incorporates unrealistic, degenerate movement patterns for UEs, which may not accurately reflect real-world dynamics and could limit the applicability of the results.
 3. Memory Constraints:
        ◦ The PTD3 algorithm has constraints on the parameter size of the Q-networks due to memory limitations. This constraint necessitates the use of smaller network architectures, potentially limiting the algorithm's performance.
    4. Coverage and Generalization:
        ◦ The performance of several state-of-the-art offline RL algorithms is erratic and does not generalize well across different datasets. This indicates a need for broader testing environments and more robust evaluation methods.
    5. Computational Expense:
        ◦ Computing the Fisher information matrix (Ft) for PTD3 is computationally expensive, requiring large batch samples and high computational resources. This could be a barrier for researchers with limited access to high-end hardware.

**Relation To Prior Work:**

Yes

**Summary And Contributions:**

The paper introduces NetworkGym, a high-fidelity network simulation environment designed to facilitate the training and evaluation of reinforcement learning (RL) algorithms for multi-access traffic management. The platform supports both online and offline RL, providing a comprehensive tool for researchers to develop and benchmark RL-based solutions for network optimization.

---

### Official Review · Reviewer_QCwj · 2024-07-29
**Interesting paper and important topic**

**Rating:** 6
**Confidence:** 4
**Correctness:** Yes
**Clarity:** Yes

**Review:**

Pros:

This paper presents a very interesting and important task. NetworkGym reduces the entry barrier for computer networking research and fosters new collaborations in the emerging field of machine learning for networking, bridging academia and industry.

The github code is well organized and easy to run.

Cons:

I feel like this paper is "half done". Although the authors mentioned in their limitation that "they assume a fixed number of UEs", but in realism this is very important to discuss and conduct some experiments to show some results on the addition or removal of UEs.

**Strengths:**

Please see Pros

**Additional Feedback:**

I'm willing to change my score if the cons are solved.

**Documentation:**

Yes

**Limitations:**

Please see Cons

**Opportunities For Improvement:**

As mentioned in the Cons, I think this paper is good enough to be considered as acceptance, but it feels like the paper is not "done" yet. If the authors can conduct some initial results on adding or removing UEs, it will be much better.

**Relation To Prior Work:**

Yes

**Summary And Contributions:**

This paper presents NetworkGym, a high-fidelity network environment simulator designed to generate multiple network traffic flows and support multi-access traffic splitting. NetworkGym enables the training and evaluation of various reinforcement learning (RL) solutions for addressing the multi-access traffic splitting problem.

---

### Official Review · Reviewer_sZTT · 2024-07-30
**2560 Review**

**Rating:** 6
**Confidence:** 3
**Correctness:** The experimental design is appropriat…

**Review:**

NetworkGym enables researchers to evaluate algorithms on a high impact real world task. The current documentation is more than sufficient to enable easy access. However, the paper could be more precise in the evaluation of the suggested algorithm.

**Strengths:**

The presented domain is of high impact to real world applications. Further, the code is well structured and documented.

**Additional Feedback:**

I have no further comments or questions.

**Clarity:**

The paper is written reasonably clearly and I found no mistakes in review. A small exception however is with regard to the code availability. From inspection it seems all the discussed features are open-source under the Apache license. The supplement states "primary code" has been released. It is unclear what primary refers to here.

The parameters alpha/beta are not defined before use, in fact they are not defined at all in the main section of the paper.

**Documentation:**

The documentation for this benchmark is excellent and extensively covers the usage and extension of the platform.

**Ethics:**

I have no ethical concerns with this work.

**Limitations:**

The authors discuss the limitations of the current state of the environment accurately and with good detail.

**Opportunities For Improvement:**

[W1] The case for NetworkGym as an offline benchmark could be stronger. From the given code it seems online algorithms were trained for the equivalent of ~2 hours of real time to a higher performance than offline methods. Some of this training time therefore would have been online-favorable as well, further shortening the real time training deficit required for online methods.

[W2] The authors make note that popular offline algorithms may under perform in new domains. It is then suggested that PTD3 is a strong contender for SOTA offline RL based on the results in a single domain.

**Relation To Prior Work:**

Related work is covered sufficiently. A more thorough discussion on how the networking environment presents different challenges for offline RL might be appropriate. However, benchmark results show that it clearly does show some other unaddressed issue for offline RL.

**Summary And Contributions:**

This paper presents NetworkGym, an environment for communication network evaluation. Experiments performed indicate current leading methods in offline RL may not transfer well to domains outside of their targeted benchmarks. An algorithm combining pessimism with TD3 is proposed and shown to be more effective at optimizing NetworkGym performance.

---

### Decision · Program_Chairs · 2024-09-26

**Decision:**

Accept (Poster)

**Comment:**

The reviewers found the paper to be of good quality and potentially can benefit the research community.